# Overshoot: Taking advantage of future gradients in momentum-based stochastic optimization

## Abstract

Overshoot is a novel, momentum-based stochastic gradient descent optimization method designed to enhance performance beyond standard and Nesterov's momentum. In conventional momentum methods, gradients from previous steps are aggregated with the gradient at current model weights before taking a step and updating the model. Rather than calculating gradient at the current model weights, *Overshoot* calculates the gradient at model weights shifted in the direction of the current momentum. This sacrifices the immediate benefit of using the gradient w.r.t. the exact model weights now, in favor of evaluating at a point, which will likely be more relevant for future updates. We show that incorporating this principle into momentum-based optimizers (SGD with momentum and Adam) results in faster convergence (saving on average at least 15% of steps). Overshoot consistently outperforms both standard and Nesterov's momentum across a wide range of tasks and integrates into popular momentum-based optimizers with zero memory and small computational overhead.

## 1. Introduction

Optimization algorithms are fundamental to machine learning. In past years, numerous SGD-like algorithms have emerged aiming to accelerate convergence, such as Adam (Kingma & Ba, 2015), RMSprop (Tieleman & Hinton., 2012), ADAGRAD (Duchi et al., 2011a), Nadam (Dozat, 2016), RAdam (Liu et al., 2020), AdamP (Heo et al., 2021) and many more. The vast majority of these algorithms utilize *momentum*, a technique that accelerates convergence in deep learning optimizers (Sutskever et al., 2013). Typically, a variation of Polyak's "classical" momentum (CM) (Polyak, 1964) or Nesterov's Accelerated Gradient (NAG)

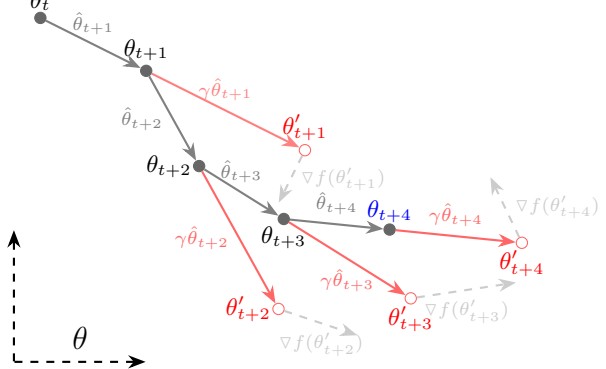

Figure 1: Overshoot derives gradients from *overshot* model weights $\theta'$, instead of from *base* weights $\theta$. The overshoot weights are "future model weights" estimations, computed by extending previous model updates by a factor of $\gamma$. This way, past gradients become more relevant to the current model weights, hence faster convergence. Consider the situation at $\theta_{t+4}$: computing the next step will use gradients coming from a more representative "neighborhood" group of overshot models (red circles) instead of a less representative "tail" of past base models (gray points).

(Nesterov, 1983) is applied. While deep learning optimization algorithms vary in many important aspects, their approach to momentum is similar: model updates are computed by aggregating the latest and past gradients. Although some optimizers, like RMSprop, do not incorporate momentum, momentum-based optimizers are frequently the default choice for optimization (Schmidt et al., 2021).

In this work, we introduce a novel approach to momentum called *Overshoot*. The primary distinction between CM and Overshoot is that in Overshoot, the gradients are computed using model weights shifted in the direction of the current momentum (future gradients). This makes Overshoot similar to NAG, however unlike NAG, Overshoot decouples the momentum coefficient and the "look-ahead" factor. To achieve that, Overshoot leverages two types of model weights: **1)** the *Base weights* ($\theta$) being optimized, and **2)** the *Overshoot weights* ($\theta'$) which are used to obtain the gradients, see Figure 1. Unlike mentioned methods, Overshoot

---

[1]Anonymous Institution, Anonymous City, Anonymous Region, Anonymous Country. Correspondence to: Anonymous Author <anon.email@domain.com>.

focuses on the process of obtaining gradients, rather than their aggregation into model updates. Consequently, the Overshoot algorithm can, in principle, be combined with any momentum-based optimization algorithm.

The contributions of this paper are two-fold:

1. The Overshoot momentum method description (section 2), complemented by efficient implementations for both the SGD and Adam[1].

2. An empirical evaluation of Overshoot across a broad range of tasks (section 5), benchmarking its effectiveness against: accelerated SGD with CM and NAG (for its simplicity) and Adam (for its widespread use).

With *rate of convergence* and *test set performance* (loss/accuracy) as metrics, our results show Overshoot beating baselines across a diverse set of scenarios. Furthermore, we implement Overshoot in SGD and Adam, achieving this with zero memory and minimal computational overheads.

## 2. Method

The Overshoot algorithm is based on the three assumptions:

**1:** In stochastic optimization, better gradient estimates lead to faster convergence and improved generalization.

**2:** The quality of the gradient estimate decreases with the $L^2$ distance between the model used to compute the gradient and the model for which the gradient is estimated. In particular, for **similar** model weights $\theta_a$ and $\theta_b$, we expect the following to be generally true:

$$||\theta_a - \theta_b|| \propto ||\nabla f(\theta_a) - \nabla f(\theta_b)|| \qquad (1)$$

where $f$ is the stochastic objective function and $\nabla f(\theta)$ denotes the vector of partial derivatives w.r.t. $\theta$. In the momentum-based optimizers, this relationship is managed by exponential weight decay scheme for past gradients.

**3:** Consecutive updates $\hat{\theta}_t$, $\hat{\theta}_{t+1}$ have similar direction:

$$\mathbb{E}\left[S_c(\hat{\theta}_t, \hat{\theta}_{t+1})\right] > 0 \qquad (2)$$

where $S_c$ is a cosine similarity. The similarity between consecutive model updates, in momentum based optimizers, is primarily determined by the momentum coefficient parameter (e.g., Adam: $\beta_1$) with its default value of 0.9 promoting update stability. Consequently, we expect the model weights, shifted in the direction of the current momentum, to be on average closer to the future model weights than the current weights are. In particular:

$$\sum_{i=0}^{2s} ||(\theta_t + s\hat{\theta}_t) - \theta_{t+i}|| < \sum_{i=0}^{2s} ||\theta_t - \theta_{t+i}|| \qquad (3)$$

---

[1]https://anonymous.4open.science/r/overshoot-47DD

where $\theta_t$ denotes model weights at step $t$, $s \in \{1, ..., K\}$ represents overshoot, and $\theta_t + s\hat{\theta}_t$ are weights shifted in the direction of the current momentum. $K$ is determined by the optimization process stability (expressed by (2)). For context: with very stable updates, $s$ times the update would take us approximately $s$ steps forward in the optimization process. By summing up to $2s$, we are effectively looking back by $s$ steps and forward by $s$ steps from that position.

Based on these premises, we **hypothesize that gradients computed on model weights shifted in the direction of the current momentum can, on average, yield more accurate future gradient estimates,** and thereby result in faster convergence. We empirically verify this hypothesis in Section 5.

Algorithm 1 describes the general form of the Overshoot algorithm. This definition is decoupled from the optimization method, and illustrates Overshoot's main idea. However, it does not achieve the desired computational and memory overheads. To do so, the Overshoot algorithm must be tailored to specific optimization methods. We discuss such implementations for SGD (SGDO) and Adam (AdamO) in Sections 2.1 and 2.2.

---

**Algorithm 1** General Overshoot definition visualized in Figure 1. We copy the optimizer function $\phi$ to address situations in which the optimizer maintains an internal state.

**input** Initial model weights $\theta_0$, stochastic objective function $f(\theta)$ with parameter $\theta$, **momentum-based** optimization method $\phi$ (e.g., Adam), learning rate $\eta > 0$, overshoot factor $\gamma \geq 0$
  $\theta_0' \leftarrow \theta_0$
  $\phi' \leftarrow \phi$
  **for** $t = 1, 2, ...$ **do**
    $g_t \leftarrow \nabla f(\theta_{t-1}')$
    $\theta_t \leftarrow \phi(\theta_{t-1}, g_t, \eta)$
    $\theta_t' \leftarrow \phi'(\theta_t, g_t, \gamma\eta)$
  **end for**
**output** $\theta_t$

---

### 2.1. Efficient implementation for SGD

The main idea of efficient implementation is to compute update vectors directly between consecutive overshoot model weights, eliminating the need for base weights and reducing computational overhead. However, without base weights, we lose the ability to reproduce its loss during training. Additionally, to fully align with the general version of Overshoot, the final overshoot weights should be updated to their base variant at the end of the training.

The update vector for Overshoot weights is computed as:

$$\hat{\theta}_{t+1}' = -\gamma\hat{\theta}_t + (\gamma + 1)\hat{\theta}_{t+1} \qquad (4)$$

where $\hat{\theta}_t$ denotes base weights update vector at step $t$. The first term: $-\gamma\hat{\theta}_t$ can be viewed as reversing the overshoot from the previous step and: $(\gamma+1)\hat{\theta}_{t+1}$ as model update from the base weights at step $t$ to the overshoot weights at step $t+1$. Using CM definition (Polyak, 1964):

$$m_{t+1} = \mu m_t + \nabla f(\theta_t) \tag{5}$$

$$\theta_{t+1} = \theta_t - \eta m_{t+1} \tag{6}$$

with $\eta > 0$ is the learning rate and $\mu \in [0,1]$ is the momentum coefficient, the Overshoot model update is:

$$\theta'_{t+1} = \theta'_t - \eta(-\gamma m_t + (\gamma+1)m_{t+1}) \tag{7}$$

By substituting the $m_{t+1}$ using CM's recurrence relation (5), and rearranging terms, (7) can be rewritten into:

$$\theta'_{t+1} = \theta'_t - \eta((\gamma - \frac{\gamma}{\mu} + 1)m_{t+1} + \frac{\gamma}{\mu}\nabla f(\theta'_t)) \tag{8}$$

In (8), update for overshoot weights is computed as a linear combination of the current momentum and gradient. Based on this observation we derive the efficient implementation of Overshoot for SGD described in Algorithm 2.

---

**Algorithm 2** SGDO: Overshoot for SGD

---

**input** Initial model weights $\theta_0$, stochastic objective function $f(\theta)$ with parameter $\theta$, learning rate $\eta > 0$, overshoot factor $\gamma \geq 0$, momentum coefficient $\mu \in (0,1]$
$\quad m_0 \leftarrow 0$
$\quad m_c \leftarrow \gamma - \gamma\mu^{-1} + 1$
$\quad g_c \leftarrow \gamma\mu^{-1}$
$\quad$ **for** $t = 1, 2, ...$ **do**
$\quad\quad g_t \leftarrow \nabla f(\theta_{t-1})$
$\quad\quad m_t \leftarrow \mu m_{t-1} + g_t$
$\quad\quad \theta_t \leftarrow \theta_{t-1} - \eta(m_c m_t + g_c g_t)$
$\quad$ **end for**
**output** $\theta_t + \eta\gamma m_t$ (Base weights)

---

## 2.2. Efficient implementation for Adam

Following the method outlined in Section 2.1, we derive an efficient implementation for Adam (AdamO) using the approach described in Equation 4. However, Adam introduces two key differences:

**1:** Momentum is defined as a decaying mean of gradients rather than a decaying sum:

$$m_{t+1} = \beta_1 m_t + (1 - \beta_1)\nabla f(\theta_t) \tag{9}$$

where $\beta_1$ denotes the momentum coefficient.

**2:** Adam applies additional operations to $m_t$ before calculating the model update $\hat{\theta}_t$, concretely $\hat{\theta}_t = \frac{\eta}{d_t}m_t$ where

$d_t = (1 - \beta_1^t)\sqrt{\hat{v}_t}$ (from Algorithm 3). The inclusion of $d_t$ complicates deriving $\hat{\theta}_t$ from momentum $m_t$, making the direct computation of (4) computationally inefficient.

There are three primary strategies to address this: (a) Compute $\hat{\theta}_t$ and $\hat{\theta}_{t+1}$ precisely at each step, leading to high computational inefficiency; (b) Cache model updates, which increases memory overhead; or (c) Use an approximation method, which we adopt here.

A key observation is that for large $t$ and with $\beta_2 = 0.999$ (resulting in small updates for the second moment estimates $v_t$ (Dozat, 2016)), the difference between $d_t$ and $d_{t+1}$ becomes small. Thus, for larger $t$ we can approximate $\hat{\theta}_t$ by applying the bias correction: $(1 - \beta_1^t)^{-1}$ and the second momentum estimate: $\sqrt{\hat{v}_t}$ from the subsequent step without significant loss in accuracy:

$$\hat{\theta}_t \approx \frac{\eta}{d_{t+1}}m_t \tag{10}$$

To avoid applying (10) at small $t$ we introduce a delayed overshoot technique to postpone the application of Overshoot. This delay is further justified by the fact that, early in training, momentum has not yet stabilized, and assumption (2) may not hold. Delayed overshoot is defined as:

$$\gamma_t = max(0, min(\gamma, t - \tau)) \tag{11}$$

where $\tau \in \mathbb{N}$ is the overshoot delay.

Starting with Equation 4, using delayed overshoot factor (11), model update estimate (10), and momentum recurrent formula (9), we derive the formula for model update used in Algorithm 3:

$$\hat{\theta}'_{t+1} = -\gamma_t\hat{\theta}_t + (\gamma_{t+1} + 1)\hat{\theta}_{t+1} \tag{4}, (11)$$

$$\approx \frac{\eta}{d_{t+1}}(-\gamma_t m_t + (\gamma_{t+1} + 1)m_{t+1}) \tag{10}$$

$$= \frac{\eta}{d_{t+1}}((-\gamma_{t+1} - \gamma_t\beta_1^{-1} + 1)m_{t+1}$$

$$+ (1 - \beta_1)\gamma_t\beta_1^{-1}\nabla f(\theta'_t)) \tag{9}$$

## 3. Related Work

In recent years, numerous new optimizers for deep learning have emerged, with many—such as RMSprop (Tieleman & Hinton., 2012), AdaGrad (Duchi et al., 2011b), Adam (Kingma & Ba, 2015), AdamP (Heo et al., 2021) AdamW (Loshchilov, 2017) RAdam (Liu et al., 2020) AdaBelief (Zhuang et al., 2020) focusing on adaptive learning schemes. These methods aim to stabilize the training process by leveraging first- and second-moment estimates to clip, normalize, and adjust gradients, facilitating smooth and stable updates.

**Algorithm 3** Adam , AdamO with good defaults: $\tau = 50$, $\gamma = 5$.

---

**input** Initial model weights $\theta_0$, stochastic objective function $f(\theta)$ with parameter $\theta$, learning rate $\eta > 0$, overshoot factor $\gamma \geq 0$, overshoot delay $\tau \geq 0$, adam betas $\beta_1, \beta_2 \in (0, 1]$
$m_0 \leftarrow 0$
$v_0 \leftarrow 0$
$\gamma_0 \leftarrow 0$
**for** $t = 1, 2, \ldots$ **do**
$\quad g_t \leftarrow \nabla f(\theta_{t-1})$
$\quad m_t \leftarrow \beta_1 m_{t-1} + (1 - \beta_1) g_t$
$\quad v_t \leftarrow \beta_2 v_{t-1} + (1 - \beta_2) g_t^2$
$\quad \hat{m}_t \leftarrow m_t (1 - \beta_1^t)^{-1}$
$\quad \gamma_t \leftarrow max(0, min(\gamma, t - \tau))$
$\quad m_c \leftarrow \gamma_t - \gamma_{t-1} \beta_1^{-1} + 1$
$\quad g_c \leftarrow (1 - \beta_1) \gamma_{t-1} \beta_1^{-1}$
$\quad \hat{m}_t \leftarrow (m_c m_t + g_c g_t)(1 - \beta_1^t)^{-1}$
$\quad \hat{v}_t \leftarrow v_t (1 - \beta_2^t)^{-1}$
$\quad \theta_t \leftarrow \theta_{t-1} - \dfrac{\eta}{\sqrt{\hat{v}_t} + \epsilon} \hat{m}_t$
**end for**
**output** $\theta_t$ $+ \gamma \dfrac{\eta}{\sqrt{v_t} + \epsilon} m_t$

---

Despite these advances, recent methods often struggle to consistently and significantly outperform Adam, which remains a reliable default choice in many scenarios (Schmidt et al., 2021) (or rather AdamW with fixed weight decay).

Other approaches, including Nadam (Dozat, 2016) and Adan (Xie et al., 2024), focus on incorporating Nesterov's accelerated gradient (NAG) into the Adam optimizer. As shown in Section 2.1, Overshoot shares similarities with Nesterov's momentum; it can be viewed as Nesterov's momentum with a parameterized "look-ahead" step used to compute gradients. Thus, Nadam and Adan also align closely with our work, albeit with distinct approaches.

Nadam employs an approximation technique to merge updates from consecutive steps into a single model update, based on the observation that for the default Adam setting ($\beta_2 = 0.999$), updates to second-moment estimates do not vary significantly. Adan, in contrast, introduces a new method called NME to precisely integrate NAG into Adam but relies on the difference between previous and current gradients, which adds memory overhead. Our approach to incorporating Overshoot into Adam is more similar to Nadam's, as it also uses an approximation method. However, unlike Nadam, we also approximate the bias-correction term after training step $t$.

The Look-ahead optimizer (Zhang et al., 2019) presents a different strategy for the optimization step, sharing some conceptual similarities with Overshoot. It also uses two sets of weights (fast and slow), but differs in that Look-ahead performs $k$ full steps with the fast weights before interpolating with the slow weights. In contrast, Overshoot applies only a single step with an increased learning rate of $k$ times and uses the resulting gradients to update the model.

In Section 4.1, we show that Overshoot unifies three distinct variants of SGD—CM, NAG, and vanilla SGD. This is similar to how a previous work (Yan et al., 2018), introduced a unification method for SGD momentum called *SUM*, which integrates Polyak's heavy ball momentum (Polyak, 1964), NAG, and vanilla SGD.

## 4. Overshoot Properties

We examine the properties of Overshoot when paired with the simplest momentum-based optimization method, SGD, and demonstrate how it unifies CM, NAG, and vanilla SGD.

### 4.1. Momentum Unification

First we examine SGDO properties, by showing its equivalence to various SGD variants. We will only focus on the Overshoot weights $\theta'$, omitting the final adjustment step to base weights.

Nesterov's accelerated gradient (NAG), can be rewritten into form of the momentum (Sutskever et al., 2013):

$$m_{t+1} = \mu m_t + \nabla f(\theta_t - \eta \mu m_t) \tag{12}$$

$$\theta_{t+1} = \theta_t - \eta m_{t+1} \tag{13}$$

SGDO update rule can be expressed as:

$$m_{t+1} = \mu m_t + \nabla f(\theta_t - \eta \gamma m_t) \tag{14}$$

$$\theta_{t+1} = \theta_t - \eta m_{t+1} \tag{15}$$

The difference between NAG and SGDO is that SGDO decouples the momentum coefficient $\mu$ and the "look-ahead" factor $\gamma$. Therefore, NAG is a special case of SGDO.

In Section 2.1, we show that the SGDO update is expressed as a linear combination of current momentum and gradient. From (8) we can see that momentum multiplication factor $(\gamma - \gamma \mu^{-1} + 1)$ can be equal to zero. In this case, the SGDO omits the momentum component, making its update rule equivalent to vanilla SGD. Taken together, SGDO (without the final adjustment step), can be equivalent to the following SGD variants:

| | |
|---|---|
| $\gamma = 0$ | SGD with CM |
| $\gamma = \mu$ | SGD with NAG |
| $\gamma = \mu(1 - \mu)^{-1}$ | vanilla SGD |

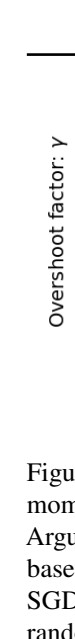

Figure 2: Overshoot for various $\gamma$ and $\mu$ settings. Negative momentum aggregates past gradients with an inverted sign. Arguably, this is not the intended behavior of a momentum based optimizer. *Estimated by minimizing (16), using SGDO to generate a series of paths with 30,000 steps and randomly sampled gradients from $\{g \in \mathbb{R}^{20} : ||g|| = 1\}$.

**4.2. Gradients relevance**

Momentum can be interpreted as a form of gradient accumulation. Its primary advantage over traditional gradient accumulation lies in its computational efficiency, as it avoids multiple forward and backward passes per optimization step. However, in momentum, gradients are computed w.r.t. historical model weights rather than the current ones. The divergence between historical and current weights reduces the relevance of historical gradients, though they remain useful to some extent. To quantify the relevance of past gradients, we introduce the measure:

$$Awd(\gamma) = \frac{1}{N} \sum_{i=1}^{N} \sum_{j=1}^{i} \left\| \theta_i - \theta'_j \right\| w(i,j) \qquad (16)$$

where $N$ is a number of training steps, $\theta_i/\theta'_i$ are base/overshoot weights at training step $i$ and $w(i,j)$ represents the weighting scheme of past gradients (accelerated SGD: $\mu^{i-j}$, Adam: $(1 - \beta_1)\beta_1^{i-j}$). Note that for non-overshoot momentum $\theta_i = \theta'_i$.

We hypothesize that $\arg\min_\gamma Awd(\gamma) > 0$, therefore Overshoot should enhance the relevance of past gradients relative to current model weights, potentially leading to faster convergence. Figure 2 illustrates the estimated $\arg\min_\gamma Awd(\gamma)$ w.r.t. $\mu$, based on simulations of the SGDO process. In Section 5.3.3 we empirically analyze $Awd$ across various tasks, comparing classical and overshoot momentum variants, and evaluate its impact on the training loss convergence rate.

**4.3. Gradient weight decay**

The exponential weight decay scheme applied to past gradients in accelerated SGD: $\mu^{i-j}$, where $i$ is the current

training step and $j$ is the step in which the gradient was computed, is suboptimal for Overshoot. This is because, in Overshoot, there is no monotonous relationship between 'gradient age' and its relevance. However, our experiments demonstrate that Overshoot outperforms CM and NAG even under the $\mu^{i-j}$ weight decay scheme (Section 5). Investigating an alternative past gradient weighting scheme, more suitable for Overshoot, is beyond the scope of this work and is left for future research.

## 5. Experiments

We evaluate the Overshoot using its efficient implementation of SGDO and AdamO, as detailed in Sections 2.1 and 2.2 using overshoot factors $\gamma \in \{3, 5, 7\}$. Overshoot results are compared against accelerated SGD, Adam, and Nadam baselines. In all experiments we use the pytorch library (2.4.0), automatic mixed precision, no learning rate schedulers, and Nvidia GPUs with Ampere architecture. AdamW weight-decay implementation is used for all Adam variants (Adam, Nadam, AdamO). We use a constant momentum coefficient in Nadam. All experiments were run with ten different random seeds, except for those in Figure 4.

**5.1. Hyper-parameters**

For baselines (CM, NAG, Adam, Nadam) and their overshoot variants we use the same set of hyperparameters. For most tasks, we use default optimization hyper-parameters:

| Variable | Name | Value |
|---|---|---|
| $B$ | Batch size | 64 |
| $lr$ | Learing rate | 0.001 |
| $\beta_1$ | Adam beta 1 | 0.9 |
| $\beta_2$ | Adam beta 2 | 0.999 |
| $\mu$ | Momentum | 0.9 |
| $\lambda$ | Weight decay | 0 |
| $\epsilon$ | Epsilon | $10^{-8}$ |

These values are either derived from the recommendations provided by the authors of the respective algorithms or set as defaults in widely used implementations. In tasks where default hyper-parameters would lead to noticeably suboptimal performance (e.g., $\eta = 0.001$ for SGD on Cifar-100), we adopt the values used in one-shot evaluations from (Schmidt et al., 2021). Any deviations from the default hyper-parameters are made to enhance the baselines and are documented in Table 1.

**5.2. Tasks**

Given that the performance of deep learning optimizers varies across tasks (Schmidt et al., 2021), we evaluate Overshoot across a diverse range of scenarios, including various

Table 1: Evaluation tasks. Configuration of 2c2d, 3c3d and VAE models are the same as in (Schneider et al., 2019). GPT-2 was fine-tuned with binary classification head, using LoRA (Hu et al., 2022).

| ID | Dataset | Model | Loss | Epochs | Parameters |
|---|---|---|---|---|---|
| MLP-CA | CA Housing Prices (Barry, 1997) | MLP with 2 hidden layers: 200, 150 | Mean squared error | 200 | - |
| VAE-FM | Fashion MNIST (Xiao et al., 2017) | Variational autoencoder (Kingma & Welling, 2013) | Mean squared error KL divergence | 100 | - |
| VAE-M | MNIST (Deng, 2012) | Variational autoencoder | Mean squared error KL divergence | 50 | - |
| 2c2d-FM | Fashion MNIST | 2 convolutional layers: 32, 64 1 hidden layers: 256 | Cross-entropy | 50 | - |
| 3c3d-C10 | CIFAR-10 (Krizhevsky, 2009) | 3 convolutional layers: 64, 96, 128 2 hidden layers: 512, 256 | Cross-entropy | 100 | $B : 128$ $lr : 0.01$ (SGD only) |
| Res-C100 | CIFAR-100 (Krizhevsky, 2009) | ResNet-18 (He et al., 2015) | Cross-entropy | 250 | $B : 256, \mu : 0.99$ $\lambda : 5e-4$ $lr : 0.01$ (SGD only) |
| GPT-GLUE | GLUE qqp (Wang et al., 2019) | GPT-2 (Radford et al., 2019) | Cross-entropy | 10 | $\lambda : 5e-4, lr : 3e-4$ |

model architectures (MLP, CNN, transformers), loss functions (Cross-entropy, Mean squared error, KL divergence), and datasets. The selection of evaluation tasks was inspired by (Schmidt et al., 2021). The detailed description of the evaluation tasks is given in Table 1. Data augmentation is only used with CIFAR-100 dataset.

### 5.3. Results

#### 5.3.1. TRAINING LOSS CONVERGENCE

Utilizing the tasks delineated in Table 1 and the methodology outlined in Section 5 we conducted empirical evaluations to compare the convergence speeds of Overshoot with those of accelerated SGD and Adam, as depicted in Figure 3. Our results indicate that Overshoot ($\gamma \in \{3, 5, 7\}$) is robust and beneficial in speeding up the convergence of the training loss. An exception was noted in the *GPT-GLUE* task, where severe over-training occurred within the initial four epochs.

To quantitatively assess the impact of Overshoot, we employed the Steps-to-95% Loss Reduction metric. Specifically, we calculate the *percentage of steps saved* to achieve 95% of the loss reduction realized by the baseline method compared to the steps used by the baseline (see Table 2).

Table 2: Percentage of training steps saved by Overshoot for various configurations.

|  | $\gamma = 3$ | $\gamma = 5$ | $\gamma = 7$ |
|---|---|---|---|
| SGD | 23.77% | 26.19% | 26.53% |
| Adam | 15.27% | 19.53% | 20.11% |

These results are averaged across all tasks (Table 1) and ten random seeds. To address the variability in mini-batch losses, we smoothed the training losses using a mean win-

dow of size 400. Reported SGDO and AdamO losses are obtained using the base model variant ($\theta_t - \gamma\hat{\theta}_t$), which we normally don't compute, as it's computationally costly and unnecessary for the optimization process.

#### 5.3.2. GENERALIZATION (MODEL PERFORMANCE)

The evaluation of performance on the test set is presented in Table 3. For most tasks, Overshoot achieves statistically significant improvements in final performance compared to both SGD and Adam baselines and never underperformed the baselines (with statistical significance). Thus, Overshoot not only proves to be advantageous but also demonstrates robustness across various tasks. However, selecting the optimal Overshoot factor $\gamma$ poses a challenge, as its efficacy varies depending on the task and the optimizer. Notably, AdamO benefits more from higher values of $\gamma$, whereas SGDO exhibits small sensitivity to changes in this parameter. The convergence of test loss is illustrated in Figure 3.

#### 5.3.3. AVERAGE WEIGHTED DISTANCE

In Section 4.2 we introduced measure to estimate relevance of past gradients used by momentum. In Figure 4 we employed AdamO for various tasks and hyper-parameters settings, to measure relation between $\gamma$, $Awd$ and loss convergence. We estimate the $Awd$ by considering past 50 models at every 50th training step, which accounts for 99.4% and 92.3% of the weighted portion of $Awd$ for $\beta_1 = 0.9/0.95$ respectively. The *GPT-GLUE* task is excluded due to its high computational and memory demands.

Empirical results confirm our hypothesis, that for default momentum coefficient $\arg\min_\gamma Awd(\gamma) > 0$, in particular

$$\arg\min_\gamma Awd(\gamma) \approx \begin{cases} 2.5, & \text{if } \beta_1 = 0.9 \\ 5, & \text{if } \beta_1 = 0.95 \end{cases}$$

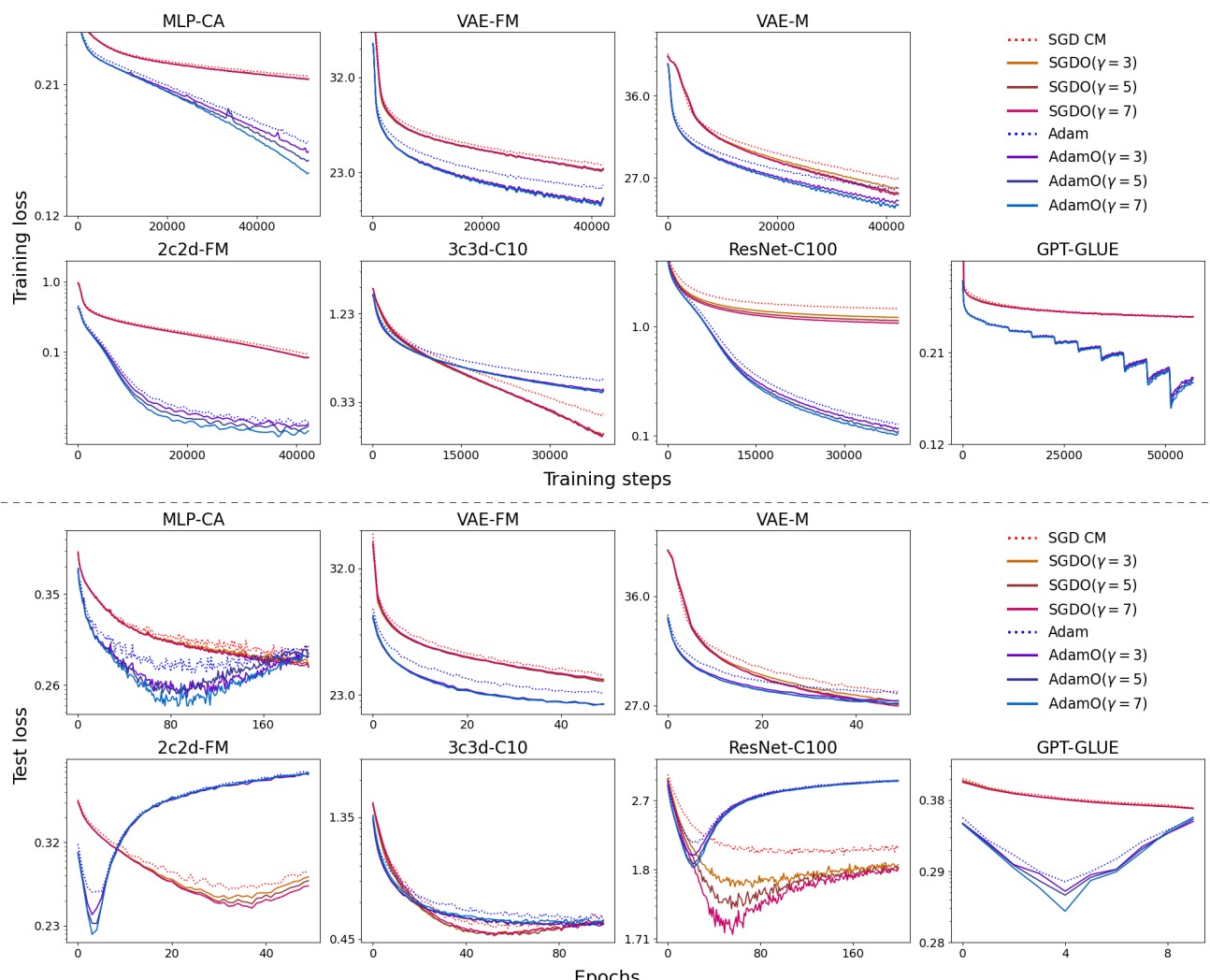

Figure 3: The average training and test losses, computed over 10 runs with different random seeds. Training losses are smoothed using a one-dimensional Gaussian filter. Obtained using the base model weights: $\theta_t - \gamma\hat{\theta}_t$. We employ a shifted logarithmic y-axis scale to visually separate small absolute differences.

However, $\arg\min_{\gamma} Awd(\gamma)$ does not necessarily yield the optimal overshoot factor for reducing training loss. We hypothesize that this discrepancy arises from the spatial distribution of overshoot weights around the base weights, which $Awd$ does not account for as a measure of distance. Further investigation into the optimal overshoot factor is warranted.

## 6. Conclusion

In this paper, we introduced *Overshoot*, a novel approach to momentum in stochastic gradient descent (SGD)-based optimization algorithms. We detailed both a general variant of Overshoot and its efficient implementations for SGD and Adam, characterized by zero memory and small com-

putational overheads. For SGD we showed that classical momentum, Nesterov's momentum and no momentum, are all special cases of the Overshoot algorithm. We evaluated Overshoot for various overshoot factors against the accelerated SGD and Adam baselines on several deep learning tasks. The empirical results suggest that Overshoot improves both the convergence of training loss and the model final performance.

## 7. Limitations

**Methodology limitations:** In this work, we only show empirical evidence of Overshoot's superiority over classical momentum and Nesterov's accelerated gradient, which could be better supported by a theoretical proof. Addition-

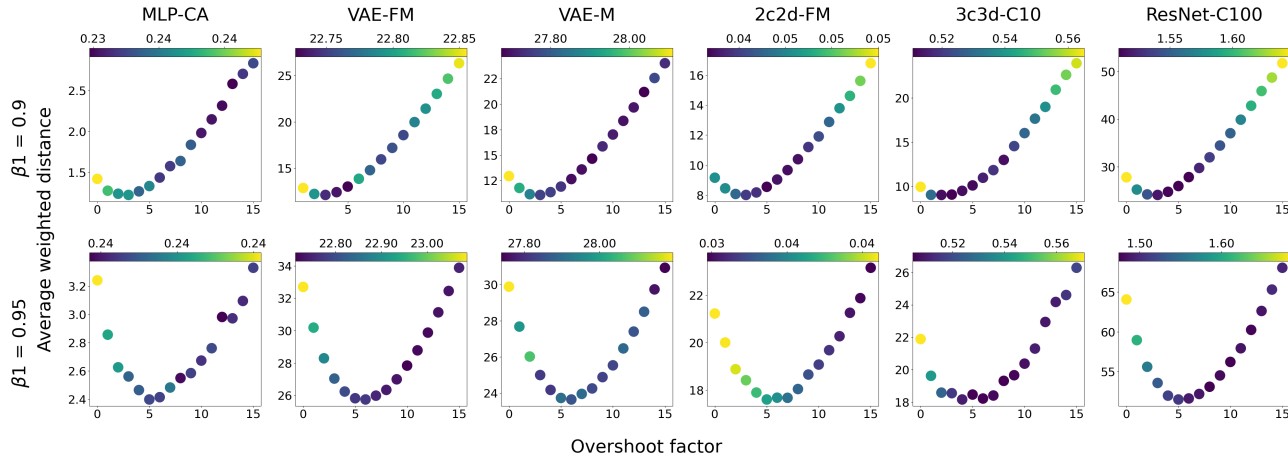

Figure 4: Relation between $Awd$ (16) and training loss (AUC) is analyzed using AdamO for $\gamma \in \{0, 1..15\}$ and $\beta_1 \in \{0.9, 0.95\}$. The training loss is visualized using a colorbar that is specific to each subgraph (lower is better). Note that AdamO with $\gamma = 0$ corresponds to the vanilla Adam optimizer. The $Awd$ is estimated by considering the distance to the past 50 model weights, sampled at every 50th training step. Training loss is computed based on the base model weights: $\theta_t - \gamma \hat{\theta}_t$. For $\beta_1 = 0.9 : \arg\min_\gamma Awd(\gamma) \approx 2.5$, and for $\beta_1 = 0.95 : \arg\min_\gamma Awd(\gamma) \approx 5$ across the tasks.

Table 3: Best achieved performance on test dataset. Evaluated after each epoch. Reporting mean values and 95% confidence interval (as subscript) from 10 runs with different random seeds. $\gamma = 3$ represents SGDO/AdamO described in Sections 2.1, 2.2 with overshot factor three. For AdamO we set the overshoot delay ($\tau$) to 50. *Statistically significant improvement over the better baseline (p-value< 0.05).

| | | SGD variants | | | | | Adam variants | | | | |
|---|---|---|---|---|---|---|---|---|---|---|---|
| | | CM | NAG | $\gamma=3$ | $\gamma=5$ | $\gamma=7$ | Adam | Nadam | $\gamma=3$ | $\gamma=5$ | $\gamma=7$ |
| Loss ↓ | MLP-CA | $26.63_{.22}$ | $26.59_{.22}$ | $26.50_{.17}$ | $26.48^*_{.16}$ | $\mathbf{26.45_{.15}}$ | $25.61_{.33}$ | $25.57_{.21}$ | $25.32^*_{.23}$ | $25.39_{.30}$ | $\mathbf{25.20^*_{.26}}$ |
| | VAE-FM | $23.39_{.10}$ | $23.28_{.06}$ | $23.29_{.07}$ | $\mathbf{23.26_{.07}}$ | $23.31_{.07}$ | $22.99_{.04}$ | $22.88_{.02}$ | $22.81^*_{.02}$ | $22.81^*_{.03}$ | $\mathbf{22.80^*_{.03}}$ |
| | VAE-M | $27.28_{.07}$ | $27.21_{.03}$ | $27.08^*_{.04}$ | $27.01^*_{.04}$ | $\mathbf{26.98^*_{.04}}$ | $27.24_{.07}$ | $27.10_{.05}$ | $27.06_{.08}$ | $\mathbf{27.01^*_{.08}}$ | $27.02^*_{.09}$ |
| Accuracy ↑ | 2c2d-FM | $92.02_{.06}$ | $92.05_{.08}$ | $92.14^*_{.08}$ | $92.20^*_{.06}$ | $92.18^*_{.07}$ | $92.20_{.13}$ | $92.23_{.15}$ | $92.36_{.08}$ | $92.37^*_{.09}$ | $\mathbf{92.43^*_{.10}}$ |
| | 3c3d-C10 | $86.14_{.20}$ | $86.46_{.15}$ | $\mathbf{86.53_{.19}}$ | $86.40_{.15}$ | $86.31_{.15}$ | $85.25_{.20}$ | $85.43_{.16}$ | $85.76^*_{.15}$ | $\mathbf{85.80^*_{.13}}$ | $85.65^*_{.16}$ |
| | Res-C100 | $52.92_{.10}$ | $53.95_{.12}$ | $55.22^*_{.12}$ | $55.64^*_{.12}$ | $\mathbf{56.07^*_{.16}}$ | $52.17_{.23}$ | $52.08_{.19}$ | $53.01^*_{.17}$ | $53.57^*_{.11}$ | $\mathbf{53.70^*_{.32}}$ |
| | GPT-GLUE | $83.27_{.11}$ | $83.18_{.10}$ | $83.36_{.04}$ | $\mathbf{83.39^*_{.08}}$ | $83.36_{.06}$ | $87.82_{.08}$ | $87.78_{.11}$ | $87.81_{.10}$ | $87.84_{.05}$ | $\mathbf{87.90_{.05}}$ |

ally, our empirical evaluations were conducted on a diverse, but limited subset of deep learning tasks. Finally, our evaluations of overshoot were conducted on default settings of optimizers without incorporating learning rate schedulers or hyperparameter fine-tuning. All of these point to the importance of future works evaluating overshoot in a wide arrange of problems and optimizer settings.

**Method limitations:** We identify two primary limitations in the presented implementation of Overshoot:

1. **The weight decay scheme for past gradients**. SGD: $\mu^{i-j}$ and Adam: $\beta_1^{i-j}(1 - \beta_1)$ where $i$ is the current training step and $j$ is the step in which the gradient was computed, prioritize the most recent gradient, which may not accurately reflect the relevance of past gradients in the Overshoot context. A revised weighting

scheme, better suited for Overshoot, could potentially enhance performance beyond the results presented.

2. **Adaptive Overshoot factor**. As demonstrated in Section 5.3.3 the optimal overshoot factor $\gamma$ varies across different tasks and hyper-parameter settings (e.g., $\beta_1$ in adam). Dynamically adjusting the overshoot factor during the training process could maximize the benefits of Overshoot. One possible method could involve monitoring (or estimating) the model update stability dynamics (expressed by (2)) and adjusting $\gamma$ accordingly.

## Impact Statement

This paper presents work whose goal is to advance the field of Machine Learning. There are many potential societal consequences of our work, none which we feel must be specifically highlighted here.

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
