# OpenReview forum: "Overshoot: Taking advantage of future gradients in momentum-based stochastic optimization"
_ICML.cc/2025/Conference — Submitted to ICML 2025_

### Official Review · Reviewer_UB8C · 2025-02-27

**Overall Recommendation:** 1

**Summary:**

Proposes a variant of Nesterov's accelerated gradient method and presents some experiments

**Claims And Evidence:**

As this paper doesn't present any theory, the strength of the paper needs to the experimental evaluation. I don't see these experiments as strongly convincing for a number of reasons:

- The biggest issue I see is the use of no learning rate schedule. This makes the comparisons meaningless as the evaluation is now far outside the regime we care about when the methods are used in practice. In addition, some methods will naturally do better than others just due to lower gradient variance or lower "effective" step size when no schedule is used.
- Since the experimental setup is non-standard, I can't determine from the reported loss values if the experiments were run correctly, or if the baselines are reasonable. Accuracy numbers of ~52-55 are very poor on c100.
- Lack of hyper-parameter tuning - tuning is necessary for a fair comparison. If using standardized setups then existing known good parameters can be used, but that is not the case here with the schedule omitted.
- Since weight decay was not used on most problems, many of the benchmarks show extreme overfitting. Weight decay changes the learning dynamics significantly, and so it's not possible to make general conclusions from results without decay.

**Essential References Not Discussed:**

N/A

**Experimental Designs Or Analyses:**

N/A

**Methods And Evaluation Criteria:**

Test problems chosen are reasonable, and multiple seeds are used.

**Other Comments Or Suggestions:**

N/A

**Other Strengths And Weaknesses:**

This paper uses non-standard notation with clashing types. For instance, m_c is used for a momentum constant, while m_t is a time varying sequence of vectors. The same letter should not be used for a constant and a vector.

Algorithm 1 is to generic, there is no reason to include this general form of the method in the paper as it's essentially so general as to be meaningless.

**Questions For Authors:**

N/A

**Relation To Broader Scientific Literature:**

Fits into an existing line of work proposing empirical modifications to momentum and averaging. Recent comparable work would be Adan.

**Theoretical Claims:**

No theory is presented in this work. The method is extremely similar to Nesterov momentum, only differing in the decoupling of one hyper-parameter, and so showing how the methods relate from a theory point of view would be interesting.

---

> ### Author Rebuttal · Authors · 2025-03-31
>
> Thank you for your evaluation and insights. We will incorporate the suggestions therein into future incarnations of our work.

---

### Official Review · Reviewer_7EGu · 2025-03-07

**Overall Recommendation:** 1

**Summary:**

This paper proposes an overshooting technique for optimization, which evaluates the gradient at point that extrapolate the standard optimizer update.

**Claims And Evidence:**

The presentation of the Algorithm is somewhat confusing: for the general Algorithm 1 it remains unclear what is allowed for the update rule $\phi$, and how $\phi'$ differs from $\phi$.
However, if we ignore Algorithm 1 for a while, the derivation of Overshoot for SGD seems to happen in Section 2.1. Using (5) in (7), the equation in (8) seems incorrect: it should be the gradient evaluated at $\theta_t$ and not $\theta_t'$. I am not sure how exactly the sequence $m_{t+1}$ is defined for the Overshoot algorithm, from Figure 1 it seems that it accumulates gradients evaluated at $\theta_t'$, and thus the derivation of (8) might reflect the intention of the authors, but needs to be fixed.

It would be highly beneficial to first define Overshoot for SGD properly with the two sequences $\theta_t$ and $\theta_t'$, and then show how it can be simplified to track only one sequence for practical implementations.

**Essential References Not Discussed:**

NA

**Experimental Designs Or Analyses:**

* It appears that the learning rate of all baseline methods (and the overshoot methods) are not tuned, but instead set to a fixed value. Given that learning rate tuning can have drastical impact on the apparent performance of a method (see for example Schmidt et al. 2021), this calls into question what can be inferred from the experimental evaluation at all.
I would recommend to compare the Overshoot methods to the baseline methods where for each method the LR is tuned via grid search. While being aware this requires massive computational effort compared to the current setup, it is the only reliable way to account for possibly different optimal learning rate for the method with/without overshooting.

* Table 2 suggests that the performance could be even better for larger $\gamma$. When does it start to degrade? Maybe larger values of $\gamma$ perform better because they use an implicitly larger learning rate (which again goes back to the point above, we can not know unless the learning rate has been tunded independently)


* It is first stated that the weight decay implementation of AdamW is used, but then weight decay is set to zero? In this case, it is not necessary to mention which weight decay implementation is used.

* Section 4.2 is missing information on which model/dataset etc. has been used to produce Figure 2. Do the insights of this section generalize for other model architectures or datasets?

**Methods And Evaluation Criteria:**

Methods are evaluated on a diverse set of optimization tasks. However, the experimental insight is limited, as learning rates are not tuned for the baseline (see details below).

**Other Comments Or Suggestions:**

Some parts of the notation are misleading/confusing/overly complicated, see details below:

* Why denote update directions as $\hat{\theta}$ when $\theta$ already denotes the actual weights? This leads to unnecessary potential of confusion.
* The coefficients $m_c$ and $g_c$ should obtain a different letter, as the current notation is in conflict with the sequences $m_t$ and $g_t$.
* Given that $\gamma_t$ is first zero, then constant at $\gamma$ it seems unnecessary to introduce this notation.
* Please refrain from using the term "weight decay scheme for gradients" (e.g. in line 092). Weight decay is an independent concept in optimization, and it is confusing to use the same term here.


Minor remarks:

* Hyperparameter table: typo in "Learning rate"
* Equation 11: max and min should be in math-operator mode and not in text mode

**Other Strengths And Weaknesses:**

The motivation for overshooting in Section 2 is relying on several strong assumptions, which are not backed up by references to prior work or theoretical arguments. For each of the assumptions made here, it should be argued why one can hope that they might be (approximately) true in practice, or if it has been reported that they hold for the relevant applications. For example, the assumption in (1) will not hold for non-smooth problems (e.g. absolute value function), as the (sub)gradient can be identical for points very far apart.

Related to this, the overshooting method also lacks theoretical insight: can the method be proven to converge under the standard assumptions (e.g. convex or Lipschitz-smooth problems)? Given the proximity to NAG and Polyak momentum, a comparative study of convergence results would be interesting.

**Questions For Authors:**

This mainly repeats the main concerns from above:

* How is the performance when all methods are reasonably tuned (in particular their step size)?
* Can you give any theoretical insight/ convergence theory for the overshoot mechanism?

**Relation To Broader Scientific Literature:**

For the motivation of the method, some more details could be provided how the Overshoot method differs from previous attempts.
In terms of theoretical comparison, this is not applicable, as no convergence results are provided.
The experimental comparison takes into account several related methods from prior work.

**Theoretical Claims:**

Please provide a proof for Equations (14)-(15). Also, is it $\theta_t$ or $\theta'_t$ in (15)?

Beyond that, the paper does not contain longer proofs that need checking.

---

> ### Author Rebuttal · Authors · 2025-03-31
>
> Thank you for putting your time into reviewing our work and for your insights. We will incorporate the suggestions therein into future incarnations of our work.
>
> > and how differs $\phi^{\prime}$ from $\phi$.
>
> $\phi$ and $\phi^{\prime}$ represents the same optimization methods, each for each weights sequence $\theta$ and $\theta^{\prime}$.
>
>
> > Using (5) in (7), the equation in (8) seems incorrect: it should be the gradient evaluated at $\theta$ and not $\theta^{\prime}$
>
> Equation 8 defines the update rule for the $\theta^{\prime}$ sequence without ever using the $\theta$ sequence. The underlying idea of this approach is described in Equation 4. In Overshoot, gradients are always evaluated at $\theta^{\prime}$. We acknowledge that adding a figure illustrating the geometric intuition behind the approach would be beneficial.
>
> > I am not sure how exactly the sequence $m_{t+1}$ is defined for the Overshoot algorithm
>
> In SGDO $m_{t+1}$ sequence is defined the same way as in CM.
>
> > Please provide a proof for Equations (14)-(15). Also, is it  $\theta^{\prime}$  or  $\theta$  in (15)?
>
> In our opinion, Equations (14) and (15) are self-evident from Figure 1 and the definition of CM (Equations 5–6). We could have used either notation,  $\theta^{\prime}$ or  $\theta$ as both describe the same principle. We chose  $\theta$ to highlight the distinction between SGDO and NAG.
>
> > Table 2 suggests that the performance could be even better for larger $\gamma$ . When does it start to degrade?
>
> This is partially addressed in Figure 4, where we tested multiple overshoot factors, not just three.
>
> > Maybe larger values of $\gamma$ perform better because they use an implicitly larger learning rate
>
> The learning rate in Overshoot does not scale with the Overshoot factor. While we understand why it might appear to do so, this is not the case. This can be seen in Figure 1 and Algorithm 2.
>
>
> >It is first stated that the weight decay implementation of AdamW is used, but then weight decay is set to zero? In this case, it is not necessary to mention which weight decay implementation is used.
>
> Weight decay is not zero across all tasks; it is applied in both Res-C100 and GPT-GLUE (see Table 1).
>
>
> > Section 4.2 is missing information on which model/dataset etc. has been used to produce Figure 2. Do the insights of this section generalize for other model architectures or datasets?
>
> Figure 2 demonstrates the equivalence of various methods with SGDO across different hyperparameter settings. Therefore, no specific dataset or model was used. The suggested optimal setting is estimated by minimizing Equation 16 using randomly generated gradients, as noted in the caption.
>
> > How is the performance when all methods are reasonably tuned (in particular their step size)?
>
> We are planning to evaluate the Overshoot method on well-tuned benchmarks in the near future. So far, we have been able to improve performance on airbench95.py (the version that does not use Muon) from https://github.com/KellerJordan/cifar10-airbench/ by approximately 10% using the Overshoot method.
>
> > Can you give any theoretical insight/ convergence theory for the overshoot mechanism?
>
> So far, we can’t offer any theoretical insights regarding convergence.

---

> > ### Comment · Reviewer_7EGu · 2025-04-09
> >
> > Dear authors,
> > thank you for the additional clarifications. In summary, providing (i) insightful benchmarks with tuned baseline methods and (ii) further motivation through theoretical results would improve this paper.

---

### Official Review · Reviewer_pFWz · 2025-03-09

**Overall Recommendation:** 1

**Summary:**

The submission presents Overshoot, a momentum optimizer that can be used with adaptive algorithms like Adam and SGD. Unlike Nesterov's Accelerated Gradient (NAG) or classical momentum (CM), Overshoot updates model weights in advance in anticipation of the upcoming momentum update even before gradients are calculated. This forward-looking approach leverages "future gradients" to provide a better estimate of future steps in optimization. The submission further presents efficient, lightweight SGD (SGDO) and Adam (AdamO) variants with effectively zero computational overhead and no additional memory. We benchmark across a set of tasks such as image classification, variational autoencoders, and GPT fine-tuning and demonstrate that Overshoot converges faster (by 15-26% fewer steps) and generalizes better than NAG, CM, and Adam.

**Claims And Evidence:**

The main benefit of the application of the proposed method is its fast convergence. while the claim of a 15% reduction in steps is based on the "Steps-to-95% Loss Reduction" metric. However, it’s not clearly defined, like, is that 95% relative to the baseline’s final loss or something else? The authors need to make it very clear to highlight your contribution.

**Essential References Not Discussed:**

The paper does not engage with more recent optimizers like AggMo and Sophia, nor with approaches that leverage adaptive look-ahead factors. Additionally, other families of optimizers—such as those designed to reduce gradient variance for faster convergence, like Katyusha, or meta-optimizers—are not discussed in the related works.

**Experimental Designs Or Analyses:**

Leaving out GPT-GLUE from parts of the analysis makes the paper’s claim of “consistent outperformance across a wide range of tasks” feel a bit shaky—especially since one of the toughest tasks (fine-tuning GPT-2) is only partially covered. It raises the question: does Overshoot really hold up when it comes to large-scale language models?

GPT-GLUE does show up in Table 2 (training steps) and Table 3 (test performance), but it’s missing from key parts of the analysis (like Awd), which makes the evaluation feel kind of piecemeal. Plus, the issue of severe over-training in GPT-GLUE (Section 5.3.1) doesn’t really get addressed in the context of Overshoot’s supposed robustness.

**Methods And Evaluation Criteria:**

The tasks cover a range of architectures (e.g., MLP, ResNet, GPT-2) and datasets (e.g., CIFAR, GLUE), but lack diversity in other important settings such as object detection and segmentation. Moreover, the problem setups appear overly simplified—for instance, training on ImageNet has become a standard benchmark for evaluating optimizer performance and should be included. Incorporating transformer-based architectures, such as ViT or Tiny ViT would also be beneficial. Additionally, using fixed hyperparameters for baseline methods may underestimate their true potential.

**Other Comments Or Suggestions:**

The problem setups appear overly simplified—for instance, training on ImageNet has become a standard benchmark for evaluating optimizer performance and should be included. Incorporating transformer-based architectures, such as ViT or Tiny ViT would also be beneficial.

**Other Strengths And Weaknesses:**

1. There are no theoretical convergence guarantees from the proposed method.
2. The tasks applied to evaluate the methods have limited difficulty and diversity.
3. Some recent and advanced optimisers are not discussed in the submission and more details can be found in the previous sections.
4. I suspect that fair comparisons are conducted to the baseline models.

**Questions For Authors:**

Some questions about the hyperparameter tuning: I think the baseline models probably are not well-tuned. Do the authors believe one set of hyperparameters for all the experiments is fair?

**Relation To Broader Scientific Literature:**

Overshoot builds on concepts such as Nesterov’s momentum and Lookahead, but distinguishes itself through parameter decoupling and single-step updates. It also connects to SUM, which seeks to unify various momentum methods.

**Theoretical Claims:**

The paper lacks rigorous convergence or gradient relevance assumption proofs. CM/NAG/SGD unification in Section 4.1 results from parameter substitution but is not formally established. However, I think these drawbacks have been mentioned in the Limitation section.

---

> ### Author Rebuttal · Authors · 2025-03-31
>
> Thank you for putting your time into reviewing our work and for your insights. We will incorporate the suggestions therein into future incarnations of our work.
>
> > while the claim of a 15% reduction in steps is based on the Steps-to-95% Loss Reduction" metric. However, it’s not clearly defined, like, is that 95% relative to the baseline’s final loss or something else?
>
> Yes, it represents the percentage of steps saved to achieve a 95% loss reduction (achieved by the baseline), compared to the baseline. We believe this metric is sufficiently explained in Section 5.3.1.
>
> > It raises the question: does Overshoot really hold up when it comes to large-scale language models?
>
> As described in Table 1, the first four tasks (MLP-CA, VAE-FM, VAE-M, 2c2d-FM) indeed share the same set of hyperparameters. The remaining three tasks (3c3d-C10, Res-C100, GPT-GLUE) use different hyperparameters to improve the performance of the baselines. We acknowledge that using underperforming baselines was not an ideal choice. However, we did so for the following reasons:
> - Limited resources for proper hyper-parameter finetuning.
> - Leave no room for potential bias in favor of the Overshoot method.
> - For the most part, we follow the one-shot setting established in previous work on benchmarking deep learning optimizers: https://arxiv.org/pdf/2007.01547
>
> Question for the reviewer: Is the problem of underperforming baselines in the experiment section the main reason for rejection?

---

### Official Review · Reviewer_poHC · 2025-03-10

**Overall Recommendation:** 2

**Summary:**

This paper draws inspiration from Nesterov’s Accelerated Gradient (NAG) and introduces a novel method called Overshoot. The Overshoot method calculates the gradient at model weights shifted in the direction of the current momentum, thereby leveraging information from the surrounding landscape more effectively. Unlike NAG, Overshoot employs a specialized reformulation that aims to reduce memory overhead.

**Claims And Evidence:**

Seems no problems.

**Essential References Not Discussed:**

None

**Experimental Designs Or Analyses:**

- The paper does not provide an adequate empirical or theoretical comparison with closely related methods. Such comparisons are crucial to highlight the advantages (or limitations) of the proposed approach relative to established  techniques.

- The experimental evaluation is constrained by the use of overly simple and outdated datasets, experimental settings, and network architectures. Consequently, the conclusions drawn from these toy scenarios lack strong evidence of generalizability to more complex, real-world tasks.

**Methods And Evaluation Criteria:**

- The core idea behind Overshoot closely resembles the well-known NAG algorithm. The discussion in the introduction does not sufficiently distinguish Overshoot from NAG.

- Moreover, through Equations (5)–(7) (and alternatively via Equations (12)–(15)),
Overshoot for SGDM is
$$
m_{t+1} = \mu m_t + \nabla f(\theta_t - \eta {\color{red}\gamma} m_t),
$$
$$
\theta_{t+1} = \theta_t - \eta m_{t+1}.
$$
While the original NAG is
$$
m_{t+1} = \mu m_t + \nabla f(\theta_t - \eta \mu m_t),
$$
$$
\theta_{t+1} = \theta_t - \eta m_{t+1}.
$$

It becomes evident that the proposed algorithm is essentially equivalent to NAG, with the only notable difference being the replacement of a coefficient. This limited distinction raises concerns about the novelty of the method. Furthermore, the practical advantages resulting from this modification remain unclear without more substantial experimental or theoretical support.

**Other Comments Or Suggestions:**

None

**Other Strengths And Weaknesses:**

see above

**Questions For Authors:**

None

**Relation To Broader Scientific Literature:**

N/A

**Theoretical Claims:**

- While the algorithm exhibits some heuristic appeal, it lacks rigorous theoretical guarantees. Additionally, the impact of the approximations introduced in AdamO requires further investigation, both theoretically and empirically, to assess their influence on convergence and performance.

---

> ### Author Rebuttal · Authors · 2025-03-31
>
> Thank you for putting your time into reviewing our work and for your insights. We will incorporate the suggestions therein into future incarnations of our work.
>
> > Unlike NAG, Overshoot employs a specialized reformulation that aims to reduce memory overhead.
>
> Overshoot does not aim to reduce memory overhead compared to standard Nesterov implementations; rather, it modifies the 'look-ahead' factor to improve convergence.
>
>
>
> > The discussion in the introduction does not sufficiently distinguish Overshoot from NAG.
>
> We distinguish between Overshoot and NAG in the introduction on line 47: ‘This makes Overshoot similar to NAG, however unlike NAG, Overshoot decouples the momentum coefficient and the ”look-ahead” factor.’
>
>
>
> > The paper does not provide an adequate empirical or theoretical comparison with closely related methods. Such comparisons are crucial to highlight the advantages (or limitations) of the proposed approach relative to established techniques.
>
> Which ‘closely related methods’ do you mean? The most relevant one is Nesterov momentum, which is used for comparison with both SGD and ADAM.
>
>
>
> > The experimental evaluation is constrained by the use of overly simple and outdated datasets, experimental settings, and network architectures. Consequently, the conclusions drawn from these toy scenarios lack strong evidence of generalizability to more complex, real-world tasks.
>
> The set of tasks was chosen based on previous work dedicated to benchmarking of deep learning optimizers: https://arxiv.org/pdf/2007.01547 What tasks would you like to be used in the evaluation?

---

### Decision · Program_Chairs · 2025-05-01

**Decision:**

Reject

**Comment:**

This paper proposes Overshoot, a novel optimization method inspired by Nesterov's Accelerated Gradient (NAG) that evaluates gradients at points extrapolated from the current optimizer update. While the paper presents an interesting modification to existing momentum-based methods, there are several significant concerns that prevent recommendation for acceptance:

The core algorithm appears to be a minor modification of NAG, primarily differing in the decoupling of one hyperparameter. The authors have not sufficiently demonstrated how this modification provides meaningful advantages over existing methods, either theoretically or empirically. The lack of theoretical analysis makes it difficult to understand the fundamental differences between Overshoot and NAG.

The empirical evaluation has several critical shortcomings: The baseline methods use fixed, untuned learning rates without proper schedules, making the comparisons potentially misleading.  The experiments omit standard benchmarks like ImageNet and modern architectures like ViT that would help establish broader applicability. Weight decay is omitted in most experiments, leading to extreme overfitting in some cases. The reported performance on some benchmarks (e.g., ~52-55% accuracy on CIFAR-100) suggests potential issues with the implementation or experimental setup.

The paper lacks theoretical analysis to support the proposed method: No convergence guarantees are provided.
Also, the assumptions made in Section 2 lack proper justification or empirical validation.

While the authors' rebuttal acknowledged these limitations and indicated plans for future work addressing some concerns (e.g., evaluating on well-tuned benchmarks), the current manuscript does not meet the standards for publication at ICML.